# Unique architecture of thermophilic archaeal virus APBV1 and its genome packaging

Denis Ptchelkine[1,2], Ashley Gillum [3], Tomohiro Mochizuki[4,5], Soizick Lucas-Staat[4], Ying Liu[4], Mart Krupovic [4], Simon E.V. Phillips [2,6], David Prangishvili[4] & Juha T. Huiskonen [3,7]

Archaeal viruses have evolved to infect hosts often thriving in extreme conditions such as high temperatures. However, there is a paucity of information on archaeal virion structures, genome packaging, and determinants of temperature resistance. The rod-shaped virus APBV1 (*Aeropyrum pernix* bacilliform virus 1) is among the most thermostable viruses known; it infects a hyperthermophile *Aeropyrum pernix*, which grows optimally at 90 °C. Here we report the structure of APBV1, determined by cryo-electron microscopy at near-atomic resolution. Tight packing of the major virion glycoprotein (VP1) is ensured by extended hydrophobic interfaces, and likely contributes to the extreme thermostability of the helical capsid. The double-stranded DNA is tightly packed in the capsid as a left-handed superhelix and held in place by the interactions with positively charged residues of VP1. The assembly is closed by specific capping structures at either end, which we propose to play a role in DNA packing and delivery.

[1] Weatherall Institute of Molecular Medicine, University of Oxford, Headley Way, Oxford OX3 9DS, UK. [2] Research Complex at Harwell, Rutherford Appleton Laboratory, Harwell OX11 0FA, UK. [3] Division of Structural Biology, Wellcome Trust Centre for Human Genetics, University of Oxford, Roosevelt Drive, Oxford OX3 7BN, UK. [4] Department of Microbiology, Institut Pasteur, 25 rue du Dr. Roux, Paris 75015, France. [5] Earth-Life Science Institute, Tokyo Institute of Technology, Tokyo 152-8550, Japan. [6] Department of Biochemistry, University of Oxford, South Parks Road, Oxford OX1 3QU, UK. [7] Helsinki Institute of Life Science and Faculty of Biological and Environmental Sciences, University of Helsinki, Viikinkaari 1, 00014 Helsinki, Finland. Correspondence and requests for materials should be addressed to S.E.V.P. (email: Simon.Phillips@rc-harwell.ac.uk) or to D.P. (email: david.prangishvili@pasteur.fr) or to J.T.H. (email: juha@strubi.ox.ac.uk)

Geothermal environments with temperatures exceeding 80 °C represent the habitat of DNA viruses with a remarkable diversity of shapes, which along with ubiquitous icosahedral and filamentous particles encompass virions resembling bottles, spindles, droplets, and coils—shapes not observed among viruses in environments with lower temperatures[1,2]. The hosts of all these viruses are hyperthermophilic members of the domain Archaea. The unusual morphotypes of hyperthermophilic archaeal viruses appear to be determined by exceptional ways of genome packaging and virion morphogenesis, which apparently also secure the stability of virions in extreme conditions of the natural environments[1]. Virion structures of the icosahedral virus STIV and the filamentous viruses SIRV2 and AFV1 of the hyperthermophilic archaeal order *Sulfolobales* have been described at high resolution[3–5]. However, many aspects determining the high thermostability of hyperthermophilic archaeal viruses remain to be elucidated.

An excellent model for such studies is the non-enveloped bacilliform virus APBV1 infecting the archaeon *Aeropyrum pernix* that has the highest optimal growth temperature, 90 °C, among aerobic hyperthermophiles[6]. The double-stranded, circular DNA genome of the virus (5.3 kbp) is among the smallest known dsDNA genomes[7]. It encodes for the major capsid protein (MCP) VP1 (8.3 kDa) in addition to three minor capsid proteins VP2 (9.7 kDa), VP3 (11.2 kDa), and VP4 (21.4 kDa). In the virion, the MCP is glycosylated[6].

To study the molecular basis of the extreme temperature resistance of APBV1, we have determined the three-dimensional structure of the purified APBV1 virion using electron cryomicroscopy (cryo-EM) to ~3-Å resolution. The structure reveals how the MCPs pack tightly together forming a tubular structure with a hydrophobic core inside the wall of the tube. The inner surface of the tube is positively charged to allow efficient interactions with the circular dsDNA genome. Together, these interactions stabilize the virus particle. The structure allows us to propose an assembly model where the dsDNA genome and capsid assemble in a concerted fashion.

## Results

**Structure of the APBV1 virion reveals helical symmetry.** APBV1 viral particles appear as rods that are 1430 Å long and 158 Å in diameter (Fig. 1a). Small punctate densities (~30 Å long) were observed protruding on the sides of the particle (Fig. 1b, c). One end of the particle appeared pointy and the other end more rounded as has been observed by electron microscopy of negatively stained specimens earlier[6]. Initial cryo-EM micrographs collected using a charge-coupled device (CCD) camera did not reveal sufficient signal for structure determination, presumably due to the small size of the major capsid protein VP1 (8.3-kDa) and the absence of prominent features on the surface of the particle to assist alignment. Subsequent data collected on a direct electron detector increased the signal-to-noise ratio and allowed further analysis (Supplementary Table 1). Particles were computationally cut into overlapping segments, which were then treated as single particles (Supplementary Table 2). Two-dimensional image classification of the segments and Fourier–Bessel analysis of power spectra calculated from 2D class averages (Supplementary Fig. 1) revealed a defined helical symmetry (rise 6.10 Å, turn 16.55 degrees, 21.75 units per turn, pitch 133.7 Å), cyclic symmetry (C5), and the polar nature of the capsid structure. The 3D density map of the particle was determined using iterative real-space helical reconstruction[8,9] to an average resolution of 3.8 Å (Supplementary Fig. 2). Local resolution analysis indicated that most of the helical capsid was resolved to 3.0 Å (Supplementary Fig. 3).

**VP1 subunits assemble via extensive hydrophopic interfaces.** In the cryo-EM map, the fold of the VP1 glycoprotein and the density for most of the large amino acid side-chains were resolved, allowing us to build a VP1 atomic model de novo and refine it (Fig. 2a, Supplementary Fig. 4, Supplementary Table 3). VP1 is almost entirely α-helical, consisting of two long α-helices (α1, residues 5–42; α2, residues 53–77) linked by a type 1 beta hairpin (residues 43–52), and flexible termini (N-term, residues 1–4; C-term, 78–81). Two adjacent prolines (Pro25 and Pro26) mediate a kink in α1, which facilitated initial assignment of the sequence to the density. α2 is also curved at Gly66, but to lesser extent than α1.

The arrangement of the VP1 subunits in the capsid resembles the blades of an impeller (Fig. 1d, e; Supplementary Fig. 4), with the small size of the VP1 protein, and more specifically the

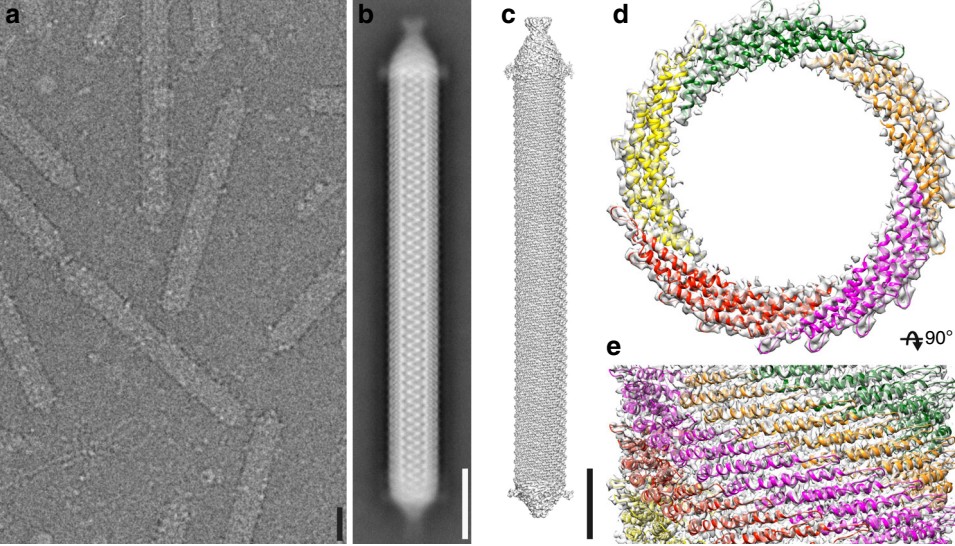

**Fig. 1** Cryo-EM and 3D reconstruction of APBV1. **a** Cryo-electron micrograph showing APBV1 viral particles in vitrous ice. Scale bar 10 nm. **b** 2D class average of the whole APBV1 particle. Scale bar 10 nm. **c** Three-dimensional model of complete APBV1 particle is shown from the side. Scale bar 10 nm. **d, e** Top (**d**) and side (**e**) view of APBV1 map (transparent surface) and model of the major capsid protein VP1. VP1 subunits related by C5 symmetry are colored in red, yellow, green, orange, and magenta

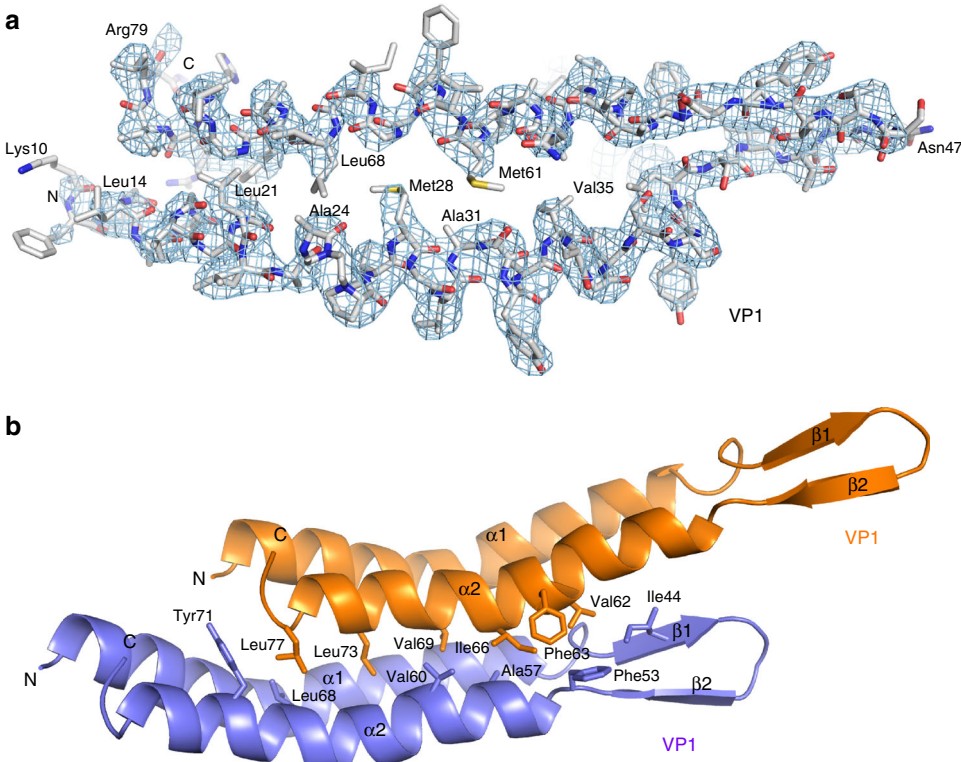

**Fig. 2** Interactions within and between VP1 subunits. **a** The cryo-EM density (mesh) and corresponding model (sticks) of VP1 are shown. In addition to the positively charged residues at the N (Lys9) and C (Arg79) termini and the putative glycosylation site at Asn47, key residues involved in intramolecular interactions are labeled. **b** Two VP1 molecules are rendered as a ribbon in orange and blue. Residues involved in intermolecular contacts between two neighboring VP1 subunits are labeled. These residues create multiple hydrophobic contacts and a large interface

curvature of the VP1 α-helices α1 and α2, allowing very compact packing of subunits. Most amino acid side-chains in both α1 and α2 helices are hydrophobic, and only the inner and outer ends of the subunits contain polar and charged residues. Each VP1 subunit makes extensive hydrophobic contacts to six other neighboring VP1 subunits, resulting in a large interface area of 2916 Å$^2$ (55.5% of the total solvent accessible area of the subunit) [10]. The total contribution to the free energy of the VP1 heptamer formation from hydophobic interactions would be 70 kcal mol$^{-1}$ according to the model proposed earlier[11,12]. A large number of aliphatic residues pack closely with each other, with Leu14, Leu21, Ala24, Met28, Ala31, and Val35 of α1 packing against Met61, Leu68, Val69, Leu73, and Ile76 of α2 to form a very tight hydrophobic interface (Fig. 2). Our analysis suggests that this interface contributes to APBV1 stability at high temperature. Interestingly, SIRV2, a rod-shaped extremophile, also uses extensive hydrophobic interfaces in its capsid assembly and stabilization[4].

**Glycosylation sequon of VP1 lies in the solvent exposed loop.** The VP1 beta hairpin is solvent accessible on the outer surface of the virus, and consists largely of polar (Ser43, Thr45, Asn47, Thr49, and Thr50) and charged (Asp52) residues. VP1 is glycosylated[6], and sequence analysis using NetNGlyc[13] revealed an N-linked glycosylation sequon (46-NST-50) located at the tip of the loop region (Supplementary Fig. 5). This sequon (NXS/T) has also been found in an archaeon *Sulfolobus acidocaldarius*[14]. This archeon and the host of APBV1 (*Aeropyrum pernix*) are both members of the same phylum, Crenarchaeota. Additional densities, preferentially located at the two ends of the helical assembly (Fig. 1b, c and Supplementary Fig. 6) could be glycans, suggesting either preferential incorporation of glycosylated VP1 subunits at

these positions or site-directed glycosylation in the context of the assembled particle. No consistent glycan density was observed at this position in the helical reconstruction, suggesting this position is only partially glycosylated. The presence of small punctate densities at the sides of individual particles suggests that a minority of the VP1 chains are also glycosylated in the helical part of the capsid, apparently at random and at a low frequency (Supplementary Fig. 1). Glycosylated peptides or other post-translational modifications were not observed in MS/MS analysis, consistent with their apparent low frequency.

**The genome of APBV1 is packaged as a left-handed superhelix.** Previous studies have shown that APBV1 genome is circular double-stranded DNA[6]. APBV1 virions assemble at extreme temperatures (90–100 °C)[6], raising a question how the viral DNA is stabilized during assembly and in the virion. Repeated experiments showed that APBV1 DNA is in circular nicked, and therefore relaxed, form, contrary to the conclusions of our previous study where the DNA band of linearized genome had been assigned as supercoiled DNA[6] (Supplementary Fig. 7). Our cryo-EM structure reveals the organization of the APBV1 DNA genome and the VP1 residues that interact with it. Five helical densities line the inner surface of the capsid, and were best resolved when the map was low-pass filtered, suggesting that they arise from a component that is less well ordered than the VP1 capsid (Fig. 3a, b). Additional density, a tube with a diameter of ~36 Å, is present on the central axis of the helical capsid (Supplementary Fig. 8). We attribute these densities to the viral genome. Although the density map does not direct assignment of the VP1–DNA contacts, or an unambiguous determination of the DNA conformation, it suggests that the DNA follows the positively charged helical tracks formed by the VP1 residues Lys9 and

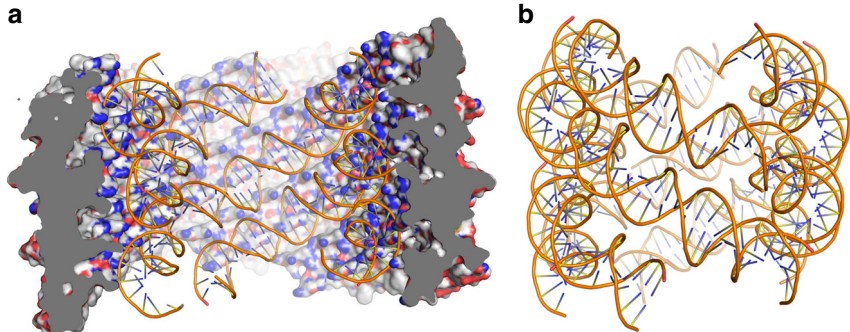

**Fig. 3** APBV1 DNA adopts the left-handed superhelix form. **a** Electrostatic surface of APBV1 is shown from inside the virion. DNA follows the positively charged tracks made up by the residues Lys9 and Arg79 of the VP1 protein. **b** Model of APBV1 DNA shows a left-handed superhelical organization. The DNA passing through the middle of the DNA superhelix was not modeled

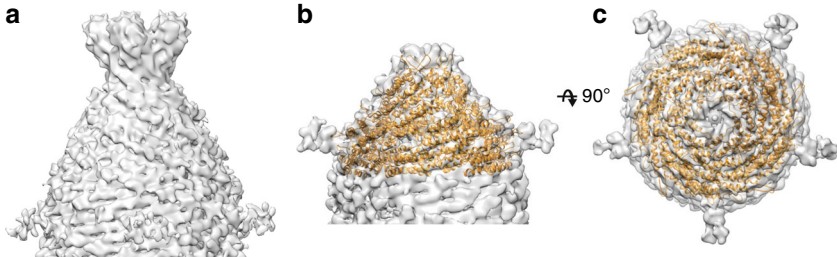

**Fig. 4** Reconstruction of the APBV1 cap structures. **a** EM map of "pointy" tip and end fibers possibly involved in recognition of the host receptor proteins. **b**, **c** EM map of "blunt" tip is shown from the side (**b**) and the top (**c**). The tip was modeled by fitting VP1 structure as a rigid body. The cap structure has five-fold symmetry. The tilt of the subunits changes with each level on approach to the tip of the cap. The additional densities on the sides of the cap were putatively assigned to a glycan attached to the VP1 subunit residue Asn47

Arg79 exposed on the inner surface of the helical capsid (Fig. 3a, b). The spacing of these basic residues along the helical tracks, is consistent with the spacing of phosphates along a B-DNA duplex. Only a fraction of DNA phosphate groups, however, makes contacts to the inner surface of the protein capsid and we predict that the DNA is likely to be fully hydrated, consistent with the B-form conformation. The curvature is, in fact, very similar to that observed in the eukaryotic nucleosome. In other helical archaeal viruses, SIRV2[4] and AFV1[5], the DNA is wrapped up by capsid proteins, and is in a relatively dehydrated A-form conformation.

The observed density, therefore, suggests that the DNA is packaged inside the capsid as a left-handed superhelix. Each of the five superhelical DNA segments that run the length of the particle would encompass a fragment of the viral genome of 808 bp according to the measured length of these densities (assuming B-form DNA). These five segments, therefore, would amount to 1238 bp less than the length of the full viral genome (5278 bp). Since the particle must package the full genome, and the circular DNA must have an even number of passes running up and down the particle, the central density observed along the axis is likely to correspond to a sixth pass of DNA (Supplementary Fig. 8). This part of the genome is not well defined in the cryo-EM map. However, it is likely to adopt a highly compact form to package all residual DNA not accounted for in the helical tracks. The polarity of the DNA must then be in the same sense in three of the helical tracks and reverse in the other two tracks and the central axis.

**Cap assemblies terminate the APBV1 capsid.** How is the helical region of the capsid capped to form the mature particle? We reconstructed 3D structures of the capsid tips using single particle

averaging and imposing five-fold symmetry (Supplementary Table 2). The two tips have distinct cap assemblies, earlier denoted as "pointy" and "blunt", and were both resolved to ~8 Å resolution. The pointy end is characterized by terminal fibers with five globular protruding domains (Fig. 4a). The terminal fibers have been shown to mediate receptor binding and cell entry in the case of the rod-shaped virus SIRV2[15]. The other end of APBV1 is devoid of fibers and has a more rounded appearance (Fig. 4b). Both tips are built from the subunits that appear structurally homologous to VP1. Minor protein VP2 shares 28% sequence identity (52% similarity) with VP1 and structure prediction[16] showed that the two are likely to have the same fold (Supplementary Fig. 5). It is thus conceivable that either or both of the caps are formed by VP2. Interestingly, only one of the two prolines (Pro25) is conserved in VP2, suggesting that the helix α1 may not be kinked the same way as in VP1. This in turn might explain the slight twist in the packing of the subunits forming the caps, when compared to VP1 in the helical region of the particle. VP2 may have evolved from VP1 by gene duplication to provide the capping function and added stability to the viral particle. While the map of the pointy tip was poorly defined, we could dock the individual subunits in the map of the blunt tip, using the VP1 model as a substitute for the unassigned tip protein (Fig. 4b, c). The structure consists of five levels of subunits related only by five-fold symmetry. While the tilt of the subunits increases, the radius of the cap reduces with each level and the last five subunits are brought together to close up the structure. Possible structural switches that transition the helical arrangement of VP1 subunits in the capsid to the more tilted arrangement of subunits in the cap regions remain to be determined in higher resolution cap structures.

## Discussion

The full model of the viral particle allows us to suggest an assembly model for APBV1. In this model, one of the caps would recognize the viral genome at three specific sites, appropriately spaced along the DNA, creating three long loops. These could intertwine under the guidance of concomitant VP1 capsid assembly, to form the observed left-handed one-in-the-middle superhelical organization in a concerted fashion. VP1 assembly into a helical capsid would proceed only together with DNA and, therefore, stop after all DNA has been covered. The open end would then be capped by minor protein(s). This assembly model makes the following predictions: (i) VP1 would be unable to form empty helical capsids in the absence of DNA, or this would be very inefficient, (ii) packaging signals should be present in the sequence, and optimally spaced by approximately twice the length of the particle, to create a virus with the observed defined length, and (iii) particles with ~2× the length should be observed if only two of the three packing signals are bound in the initiation of packaging, leading to an assembly where four of the five helical tracks are occupied (and 3× if only one of the three packing signals is bound and two of the five helical tracks are occupied). In our initial search of the DNA sequence, obvious putative packaging signals were not evident and there may not be enough data to find a consensus sequence. Consistent with our assembly model, empty particles devoid of DNA were not observed in our preparations and double length particles are occasionally observed[6].

In conclusion, the tight hydrophobic core of the VP1 layer and extensive Coulombic interactions between VP1 termini and the negatively supercoiled DNA genome facilitate assembly of the ABPV1 particle with extreme thermostability. The extent of these contacts and the fact that empty particles were not observed points to a concerted assembly model where VP1 and genome condense together. The exact molecular interactions in the capping structures and the interaction of the virion with its archaeal host remain to be determined in further studies.

## Methods

**Virus propagation and purification**. APBV1[6] was propagated in *A. pernix* strain K1[17]. Virions were collected from cell-free supernatant by precipitation with polyethylene glycol and purified by CsCl density gradient centrifugation in a Beckman SW60 rotor at 48,000 rpm for 24 h. The virion-containing fractions were collected and dialyzed against distilled water.

**Determination of genome superhelical state**. DNA was isolated from purified APBV1 virions following a slightly modified phenol–chloroform extraction protocol[18] as described earlier[6]. The viral DNA genome was treated with different concentrations of mung bean nuclease (MBN-1: 30 U µg$^{-1}$ DNA, MBN-2: 20 U µg$^{-1}$ DNA, and MBN-3: 10 U µg$^{-1}$ DNA) and incubated for 30 min at 30 °C. The DNA digestion products were separated on 1% agarose gel and identified with SYBR Safe stain.

**Electron cryomicroscopy**. A 3-µl aliquot of sample was applied to a glow-discharged grid (C-flat; Protochips, Raleigh, NC) at ambient temperature and ~80% relative humidity and vitrified by plunge-freezing into liquid ethane using a vitrification apparatus (CP3, Gatan, Pleasanton, CA). Data were acquired using a 300-kV transmission electron microscope (Tecnai F30 'Polara'; FEI) equipped with an energy filter (slit width 20 eV; GIF Quantum, Gatan) and a direct electron detector (K2 Summit, Gatan). Movies (22 frames, each frame 0.2 s) were collected in electron counting mode at dose rate of 8 e$^{-}$ pixel$^{-1}$ s$^{-1}$ at calibrated magnification of 37,037× resulting in a total dose of ~22 e$^{-}$ Å$^{-2}$ and pixel size of 1.35 Å.

Movie frames were aligned to account for drift using Motioncorr[19] and contrast transfer function (CTF) parameters were estimated using CTFFIND3[20]. A total of 400 drift-corrected micrographs were used to pick 3145 virus particles using e2helixboxer.py from EMAN2. Helical reconstruction was calculated in Spring[9]. Particles were segmented into overlapping segments, which were treated as single particles. Initial helical parameters were derived by layer-line analysis of 2D class averages to assign layer-line heights and Bessel orders. The parameters were refined by calculating amplitude correlation for a set of different parameters around the original ones[21]. The final map was calculated using iterative real-space gold-standard refinement.

The ends of the virus particle were reconstructed using Relion. An initial estimate for the in-plane rotation (psi) was calculated from the original coordinates of the helix ends and all filament ends were subjected to 2D classification in a template-free manner. Centering of the helix ends was improved by manually defining the particle centers in the class averages, which allowed us to calculate more accurate particle coordinates in the micrographs and re-extract them for a second round of 2D classification. The 2D classification separated the filament ends into a set of pointed ends and a set of rounded ends. For 3D refinement of particles in both sets, initial models were calculated using *relion_reconstruct*. First, out-of-plane rotation (tilt) was set to 90° and the angle around the long axis of the filament (rot) was set to a random value. Gold standard local refinement was run for both sets restricting deviations from the initial psi and tilt angles by sigma value of 2° and 4°, respectively. Initial refinements run without applying symmetry confirmed the presence of C5 symmetry in both ends and C5 symmetry was applied in subsequent refinements.

Resolution of the models was estimated by using Fourier shell correlation in Relion using 0.143-cutoff and phase randomization to account for masking effects[22]. Local resolution was estimated using ResMap[23]. Reconstruction statistics are summarized in Supplementary Table 2.

A 2D class average of the complete particle was created in Relion. Extracted particles were downsampled by factor of 2× resulting in a box size of 700 pixels and pixel size of 2.7 Å pixel$^{-1}$. The final average of particles in the best class was calculated in *relion_reconstruct* up to 1/8 Å$^{-1}$ spatial frequency.

**Atomic model building and refinement**. An atomic model for VP1 in the helical part of the filament was traced with Buccaneer[24], built with Coot[25], and refined in Refmac[26] and Phenix[27] applying secondary structure, rotamer, and Ramachandran plot restraints. The ends of the filament were modeled in COOT by jelly-body fitting several copies of the VP1 atomic model. The refinement and geometry of the models were validated with Refmac[26] and MOLPROBITY[28]. Model refinement statistics are summarized in Supplementary Table 3.

**Data availability**. Density maps and atomic models reported in this paper have been deposited in the Electron Microscopy Database and in the Protein Databank under accession codes EMD-3857 (helical capsid reconstruction), EMD-3858 (blunt end reconstruction), EMD-3859 (pointy end reconstruction), and PDB:5OXE (VP1 atomic model). The authors declare that all other data supporting the findings of this study are available within the article and its Supplementary Information files, or are available from the authors upon request.

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

## Acknowledgements

The OPIC electron microscopy facility was founded by a Wellcome Trust JIF award (060208/Z/00/Z) and is supported by a WT equipment grant (093305/Z/10/Z). Work in D.P.'s laboratory is supported by l'Agence Nationale de la Recherche (France) and by the European Union's Horizon 2020 research and innovation program under grant agreement 685778, project VIRUS-X. Work in J.T.H.'s laboratory is supported by European Research Council (649053) under the European Union's Horizon 2020 research and innovation program. The Wellcome Trust Centre for Human Genetics is supported by a Wellcome Trust Core Award (203141/Z/16/Z).

## Author contributions

D.Pr., S.E.V.P. and J.T.H. designed the study. D.Pt., A.G. and J.T.H. performed cryo-EM data collection. D.Pt. and J.T.H. reconstructed EM maps. D.Pt. built and refined atomic models. T.M., S.L.-S. and Y.L. optimized the procedure, purified APBV1 virus, and performed genome folding-assessment assays. M.K. performed sequence analysis. D.Pt., S.E.V.P., D.Pr. and J.T.H. wrote and all authors commented on the manuscript.

## Additional information

**Competing interests:** The authors declare no competing financial interests.

