## [Peer Review File · Nature Communications]

Reviewers' comments:

Reviewer #1 (Remarks to the Author):

The manuscript "Unique architecture of thermophilic archaeal virus APBV1 and its genome packaging" by Ptchelkine et al describes the cryoEM structure of the hyperthermophilic archaeal virus APBV1. The overall quality of the map is very high, and the density for the helical central region was ~ 3 Å resolution, allowing the authors to build the first atomic structure of the APBV1 major capsid protein. The two slightly different end-caps of the virus were of lower resolution (~ 7 Å each) but allowed recognition of the fold of the minor capsid which resemble the major capsid fold. Of particular interest, the authors were able to visualize the circular dsDNA genome organized as a left-handed DNA superhelix that followed the helical periodicity of the internal capsid. Further, the authors propose that a central tube of density within the superhelix is the last part of the genome, and that by threading the DNA in this central tube area the virus is able to accommodate the full genome. The paper is clearly written, and makes compelling arguments regarding capsid stabilization in extreme environments, and the molecular basis for an unusual arrangement of dsDNA. Thus, the manuscript should be of general interest to Nature Communications readers, and of particular interest to those interested in thermophiles, virus structure/geometry, and DNA structure under confinement. The only minor suggestions this reviewer has are:

- 1) A brief discussion of the transition area between end-caps and central region would be interesting, especially if the authors have an idea about the molecular nature of how the structure changes at this transition points.
- 2) Call me old-fashioned, but stereo-diagrams for figures 2, 3, and 4 would help the reader better visualize this remarkable structure.
- 3) It might be worth taking another look at the problematic regions of the structure. 88% in the favored region of the Ramachandran plot is a little low (should be mid 90s), and 3 Å resolution isn't good enough to justify residues with poor geometry.

Reviewer #2 (Remarks to the Author):

This manuscript by Ptchelkine et al. present a near-atomic resolution cryo-EM structure of a rod-shaped *Aeropyrum pernix* bacilliform virus 1 (APBV1) infecting *Aeropyrum pernix*, a hyperthermophilic archaea that grows at extremely high temperature (above 80 Celsius degree). This work provides the following valuable information: 1) The 3.0 angstrom resolution structure of the major capsid protein (VP1 glycoprotein) shows that the hydrophobic interaction between the alpha-helices in the VP1 subunits contributes to the stability of the capsid. 2) The glycans attached to the VP1 subunits were observed in the capsid. 3) A part of the viral genome that follows the symmetry of the capsid was resolved, showing the electrostatic interactions between the major capsid protein and the genome. Based on these observations, the authors proposed a reasonable model for APBV1

assembly. The authors attempted to “study the origin of the extreme temperature resistance of APBV1, and the molecular architecture underlying its unusual life cycle” and attributed the extreme thermostability of the APBV1 to the hydrophobic interaction between the VP1 subunits and extensive Coulombic interactions between VP1 termini and the DNA genome. However, the authors did not compare the structural similarities and differences between the APBV1 and other hyperthermophilic archaea viruses or viruses growing at normal temperatures.

I wonder why the authors did not include the line and page numbers.

Major points:

The introduction section is overly succinct. A more detailed summary of the unique structural features of other archaea viruses is needed (see Dellas et al., *Annu. Rev. Virol.* 2014. 1:399–426). A more detailed summary of the unique structural features of other archaea viruses is needed (see Dellas et al., *Annu. Rev. Virol.* 2014. 1:399–426). A more detailed summary of the unique structural features of other archaea viruses is needed (see Dellas et al., *Annu. Rev. Virol.* 2014. 1:399–426).

Page 4, Lines 18-19: Comparisons with other rod-shaped viruses optimally at extremely high temperatures and normal temperatures are needed to support this conclusion.

Page 5, line 17: The DNA double helix structure should be visible at the 8 angstrom resolution. A figure fitting the dsDNA models to the densities is needed, possible to be included in the supplementary information.

Page 6, line 17: Why is the C5 symmetry imposed here? Considering that symmetry-mismatch organization is ubiquitous in viral structure, the C5 cyclic symmetry does not mean that the cap is C5 symmetrical.

Page 6, line 23: VP2 should be introduced in the Introduction. What is the amino acid sequence identity between VP1 and VP2?

Page 7, line 11: A brief description of virus propagation and purification is needed here.

Fig. 4a: The five globular domains are unclear in Fig. 4a.

Minor points:

Page 2, line 3: Aeroperum appears to be a typo for Aeropyrum. This typo appears a number of times throughout the manuscript.

Page 4, last line: “S. acidocaldarius” is redundant.

Supplementary Fig. S1: Add a figure legend for each panel.

Supplementary Fig. S5: Show the location of the zoom-in view in the overall view of the capsid structure.

Supplementary Fig. S7: Scale bars are needed.

Reviewers' comments:

Reviewer #1 (Remarks to the Author):

The manuscript "Unique architecture of thermophilic archaeal virus APBV1 and its genome packaging" by Ptchelkine et al describes the cryoEM structure of the hyperthermophilic archaeal virus APBV1. The overall quality of the map is very high, and the density for the helical central region was ~ 3 Å resolution, allowing the authors to build the first atomic structure of the APBV1 major capsid protein. The two slightly different end-caps of the virus were of lower resolution (~ 7Å each) but allowed recognition of the fold of the minor capsid which resemble the major capsid fold. Of particular interest, the authors were able to visualize the circular dsDNA genome organized as a left-handed DNA superhelix that followed the helical periodicity of the internal capsid. Further, the authors propose that a central tube of density within the superhelix is the last part of the genome, and that by threading the DNA in this central tube area the virus is able to accommodate the full genome. The paper is clearly written, and makes compelling arguments regarding capsid stabilization in extreme environments, and the molecular basis for an unusual arrangement of dsDNA. Thus, the manuscript should be of general interest to Nature Communications readers, and of particular interest to those interested in thermophiles, virus structure/geometry, and DNA structure under confinement.

We would like to thank the reviewer for their accurate review of our work and positive comments.

The only minor suggestions this reviewer has are:

1) A brief discussion of the transition area between end-caps and central region would be interesting, especially if the authors have an idea about the molecular nature of how the structure changes at this transition points.

This is an interesting point. In the original version of the manuscript we already discuss the possible role of VP2 Pro25 in transitioning to a different curvature.

In the absence of further evidence the following sentence has been added (page 8, lines 9–11): "Possible structural switches that transition the helical arrangement of VP1 subunits in the capsid to the more tilted arrangement of subunits in the cap regions remain to be determined in higher resolution cap structures."

2) Call me old-fashioned, but stereo-diagrams for figures 2, 3, and 4 would help the reader better visualize this remarkable structure.

We would like to point out that the maps will be deposited in the EMDB and PDB and various online tools exist for interactive 3D visualization. In this light we have decided to not include stereo-diagrams.

3) It might be worth taking another look at the problematic regions of the structure. 88% in the favored region of the Ramachandran plot is a little low (should be mid 90s), and 3 Å resolution isn't good enough to justify residues with poor geometry.

We would like to thank the reviewer for the opportunity to improve our VP1 model geometry. We have now carefully re-examined the structure and 94.1% of the peptide bonds are now in the favoured region. Supplementary Table 3 has been

updated accordingly and PDB validation report has been included as part of the submission.

Reviewer #2 (Remarks to the Author):

This manuscript by Ptchelkine et al. present a near-atomic resolution cryo-EM structure of a rod-shaped Aeropyrum pernix bacilliform virus 1 (APBV1) infecting Aeropyrum pernix, a hyperthermophilic archaea that grows at extremely high temperature (above 80 Celsius degree). This work provides the following valuable information: 1) The 3.0 angstrom resolution structure of the major capsid protein (VP1 glycoprotein) shows that the hydrophobic interaction between the alpha-helices in the VP1 subunits contributes to the stability of the capsid. 2) The glycans attached to the VP1 subunits were observed in the capsid. 3) A part of the viral genome that follows the symmetry of the capsid was resolved, showing the electrostatic interactions between the major capsid protein and the genome. Based on these observations, the authors proposed a reasonable model for APBV1 assembly. The authors attempted to “study the origin of the extreme temperature resistance of APBV1, and the molecular architecture underlying its unusual life cycle” and attributed the extreme thermostability of the APBV1 to the hydrophobic interaction between the VP1 subunits and extensive Coulombic interactions between VP1 termini and the DNA genome. However, the authors did not compare the structural similarities and differences between the APBV1 and other hyperthermophilic archaea viruses or viruses growing at normal temperatures.

We would like to thank the Reviewer for their comments and constructive suggestions that we address below.

I wonder why the authors did not include the line and page numbers.

We apologise for this omission and have now added line and page numbers.

Major points:

The introduction section is overly succinct. A more detailed summary of the unique structural features of other archaea viruses is needed (see Dellas et al., Annu. Rev. Virol. 2014. 1:399–426). [duplicate text removed]

We have extended Introduction and added the reference.

Page 4, Lines 18-19: Comparisons with other rod-shaped viruses optimally at extremely high temperatures and normal temperatures are needed to support this conclusion.

We have softened our conclusion and added comparisons to other rod-shaped viruses growing at high-temperatures (page 5, lines 13–15): “Our analysis suggests this interface contributes to APBV1 stability at high temperature. Interestingly, SIRV2, a rod-shaped extremophile, also uses extensive hydrophobic interfaces in its capsid assembly [DiMaio et al. 2015]”, and (page 7, lines 1–2): “In other helical archaeal viruses, SIRV2 [DiMaio et al. 2015] and AFV1 [Kasson et al. 2017], the DNA is wrapped up by capsid proteins, and is in a relatively dehydrated A-form conformation.

Page 5, line 17: The DNA double helix structure should be visible at the 8 angstrom resolution.

The DNA double helix is not resolved well enough to observe the minor groove. Due to the inherent disorder, we have refrained giving a resolution estimate. We originally wrote: “helical densities ... were best resolved when the map was low-pass filtered to resolution of 8 Å or lower”. To avoid possible confusion, the text has been modified to state (page 6, lines 15–16): “helical densities ... were best resolved when the map was low-pass filtered”.

A figure fitting the dsDNA models to the densities is needed, possible to be included in the supplementary information.

As suggested, we have modified Supplementary Figure 7 to show the fitting of the DNA model to the density.

Page 6, line 17: Why is the C5 symmetry imposed here? Considering that symmetry-mismatch organization is ubiquitous in viral structure, the C5 cyclic symmetry does not mean that the cap is C5 symmetrical.

We agree and had confirmed the presence of C5 symmetry computationally. We have modified the text to add this detail (page 10, lines 22–24): “Initial refinements run without applying symmetry confirmed the presence of C5 symmetry in both ends and C5 symmetry was applied in subsequent refinements.”

Page 6, line 23: VP2 should be introduced in the Introduction.

We have added VP2 in the Introduction (page 3, lines 16–18): “[the genome] encodes for the major capsid protein (MCP) VP1 (8.3 kDa) in addition to three minor capsid proteins VP2 (9.7 kDa), VP3 (11.2 kDa), and VP4 (21.4 kDa)”.

What is the amino acid sequence identity between VP1 and VP2?

The following text has been added (page 7, lines 23–24): “Minor protein VP2 shares 28% sequence identity (52% similarity) with VP1”.

Page 7, line 11: A brief description of virus propagation and purification is needed here.

The following text has been added (page 9, lines 12–15): “APBV1 virus was produced and purified as described earlier [Mochizuki et al. 2010]. Shortly, APBV1 was propagated in *A. pernix* strain K1. Virions were collected from cell-free supernatant by precipitation with polyethylene glycol, and purified by CsCl density gradient centrifugation in a Beckman SW60 rotor at 48,000 rpm for 24 h. The virion-containing fractions were collected and dialyzed against distilled water.”

Fig. 4a: The five globular domains are unclear in Fig. 4a.

We have replaced the panel with one showing a larger area of the map. The globular domains are now shown.

Minor points:

Page 2, line 3: Aeroperum appears to be a typo for Aeropyrum. This typo appears a number of times throughout the manuscript.

This typo has been fixed.

Page 4, last line: "S. acidocaldarius" is redundant.

The text has been modified as follows (page 5, lines 22–23): "This archeon and the host of APBV1 (*Aeropyrum pernix*) are both members of the same phylum, Crenarchaeota."

Supplementary Fig. S1: Add a figure legend for each panel.

Figure S1 does not consist of multiple panels as such, the separate images are examples of 2D class averages and the same legend applies to all of them as a whole.

Supplementary Fig. S5: Show the location of the zoom-in view in the overall view of the capsid structure.

The figure has been modified as suggested.

Supplementary Fig. S7: Scale bars are needed.

We are now showing the panels in the same scale and have added a DNA model that provides the scale.

REVIEWERS' COMMENTS:

Reviewer #1 (Remarks to the Author):

The authors addressed the two most important issues raised by this referee. I don't agree that depositing the map in the EMDB justifies not having a stereo-image, as this requires considerable effort on the part of the reader. However, that is a fairly trivial issue. Overall, it is a very exciting manuscript!

Reviewer #2 (Remarks to the Author):

The authors have addressed all my comments. I am satisfied with their responses and changes.

REVIEWERS' COMMENTS:

Reviewer #1 (Remarks to the Author):

The authors addressed the two most important issues raised by this referee. I don't agree that depositing the map in the EMDB justifies not having a stereo-image, as this requires considerable effort on the part of the reader. However, that is a fairly trivial issue. Overall, it is a very exciting manuscript!

We would like to thank the reviewer for their positive comment. Stereo images of the VP1 atomic structure and a portion of the map have now been included as a new supplementary figure (Fig S4). A stereo image for the DNA model has been added to old supplementary figure (Fig S8).

Reviewer #2 (Remarks to the Author):

The authors have addressed all my comments. I am satisfied with their responses and changes.

We would like to thank the reviewer for reviewing our revised manuscript.